# MODALS: Modality-agnostic Automated Data Augmentation in the Latent Space

**Tsz-Him Cheung & Dit-Yan Yeung**
Department of Computer Science and Engineering
The Hong Kong University of Science and Technology
{thcheungae,dyyeung}@cse.ust.hk

## Abstract

Data augmentation is an efficient way to expand a training dataset by creating additional artificial data. While data augmentation is found to be effective in improving the generalization capabilities of models for various machine learning tasks, the underlying augmentation methods are usually manually designed and carefully evaluated for each data modality separately. These include image processing functions for image data and word-replacing rules for text data. In this work, we propose an automated data augmentation approach called MODALS (Modality-agnostic Automated Data Augmentation in the Latent Space) to augment data for any modality in a generic way. MODALS exploits automated data augmentation to fine-tune four universal data transformation operations in the latent space to adapt the transform to data of different modalities. Through comprehensive experiments, we demonstrate the effectiveness of MODALS on multiple datasets for text, tabular, time-series and image modalities.[1]

## 1 Introduction

Deep learning models tend to perform better with more labeled training data. However, labeled data are usually scarce and expensive to collect. Data augmentation is a promising means to extend the training dataset with new artificial data. In image recognition, image processing functions, like randomized cropping, horizontal flipping, and color shifting, are commonly adopted in modern image recognition models (Krizhevsky et al., 2012; Shorten & Khoshgoftaar, 2019). Following the success of image augmentation, it is becoming increasingly common to apply data augmentation in natural language processing tasks, like machine translation, text classification, and semantic parsing. Various word-based transformations have been proposed to perturb word tokens, such as replacing similar words or phrases, swapping word orders, and inserting or dropping random words (Cheng et al., 2018; Şahin & Steedman, 2018; Wei & Zou, 2019).

Over the years, more transformation functions have been proposed to augment different datasets. Cutout randomly occludes a part of an image to avoid overfitting (Devries & Taylor, 2017b). For label-mixing methods, CutMix replaces the occluded part in Cutout by a different image (Yun et al., 2019) and Mixup interpolates two images with their corresponding one-hot encoded labels (Zhang et al., 2018). These methods have been tested and found to be effective in multiple image datasets. Alternatively, new data can be created using deep generative models, for example, using GAN-based approaches to generate new images (Antoniou et al., 2017; Sandfort et al., 2019), conditional pre-trained language models to generate training sentences (Kumar et al., 2020), and back-translation to paraphrase sentences by translating sentences to another language and back to the original language (Xie et al., 2020). While these generative approaches are found to be useful, the generators or language models are often hard to implement and are expensive to train. Apart from advancing individual transformations, another line of research studies their optimal composition. As the choice and order of the transformations are decided and tested manually, the success of an augmentation scheme in one dataset may not generalize well to other datasets. To tackle this problem, AutoAugment as an automated data augmentation method was proposed to automate this process by learning

---

[1]Code is available at https://github.com/jamestszhim/modals.

an optimal augmentation policy, which decides the probability and magnitude to apply pre-defined transformations (Cubuk et al., 2019).

Whether it is for standard augmentation or automated augmentation, the transformation functions are often designed and tested carefully for each data modality separately. While it may be intuitive to create new and valid images using image processing techniques, it is non-trivial to define such label-preserving transformations on discrete data like text data. This prohibits the reuse of augmentation schemes across different data modalities. Beyond supervised learning, there is an increasing trend of utilizing data augmentation to extract information from unlabelled data in unsupervised (Xie et al., 2020) or self-supervised learning tasks (Chen et al., 2020; Grill et al., 2020). These current methods are heavily dependent on existing augmentations in vision applications. In order to generalize these methods to other modalities like text and graph data, there is a need for robust data augmentation for each modality. Therefore, we propose MODALS to apply modality-agnostic automated data augmentation in the latent space. The idea of transforming latent features is inspired by representation learning. For image generation, the work of Upchurch et al. (2017) interpolates images along specific directions in the latent space to add new semantics without changing the class identity, such as adding facial hair to the image of a male face by translating the corresponding latent representation towards the direction of male faces with facial hair. This suggests that augmenting data in the latent space can capture diverse semantic transformations which are usually hard to define in the input space.

Augmentation in the latent space poses two main challenges: learning a latent space that is continuous for transformation, and finding the effective directions to traverse. Failing to address them properly may cause an augmented example to lose its original class identity. In the previous work by Devries & Taylor (2017a), the latent space is learned by training an autoencoder, which encodes the input data into a latent vector and decodes it back to the original data. The learned latent representations are then transformed by interpolation, extrapolation, or adding Gaussian noise and decoded as synthetic examples for the downstream tasks. For image data, ISDA estimates the semantic directions by inspecting the feature covariance (Wang et al., 2019). LSI and Manifold Mixup apply Mixup to the latent feature vectors (Liu et al., 2018; Verma et al., 2019).

We develop our framework as MODALS. At a high level, MODALS applies latent space augmentation to address the data augmentation problem in multiple data modalities. Instead of improving the model performance in a specific domain or modality, the major focus and novelty of this work is to propose a general automated data augmentation framework that can work for multiple data modalities in a generic way. To the best of our knowledge, such an attempt has not been made by others in the research community. MODALS also differs from the previous approaches in three ways. First, as opposed to other operation-based latent space transformation methods, MODALS is jointly trained with augmentation. As it involves no auxiliary models or additional processes to generate examples, it can be efficiently integrated into the popular deep learning frameworks. Second, we observe that examples that are more uncertain to predict, or considered to be hard in the active learning literature, tend to carry richer information for model training. Therefore, we modify standard latent space transformations to create harder examples in MODALS. Third, MODALS introduces additional loss terms to improve the quality of augmentation in the latent space. In summary, we make four major contributions in this paper:

- Propose a framework to apply automated augmentation in the latent space.
- Propose a novel and effective way to create hard examples.
- Study additional loss terms to improve label-preserving transformations in the latent space.
- Evaluate MODALS extensively on classification datasets across multiple data modalities using various deep learning models.

## 2 RELATED WORK

### 2.1 AUTOMATED DATA AUGMENTATION

In practice, multiple transformations are used compositely to augment a dataset. The choice and strength of the transformations affect the model performance on different datasets. Motivated by neural architecture search and reinforcement learning, automated data augmentation formulates the

tuning of augmentation parameters as a learning task. AutoAugment trains a child model with the augmentation transformations generated by a controller, which is updated by reinforcement learning using the downstream validation accuracy as the reward signals (Cubuk et al., 2019). To reduce the search effort in AutoAugment, various approaches have been proposed. Population Based Augmentation (PBA) adopts an evolution-based search strategy to simultaneously train and perturb a policy with multiple child models (Ho et al., 2019). RandAugment replaces the augmentation parameters in AutoAugment by uniform values across all transformations and searches for the magnitude and the number of transformations to apply (Cubuk et al., 2020). Adversarial AutoAugment finds an adversarial policy that creates difficult examples (Zhang et al., 2020). Luo et al. (2020) considered a text image recognition task and trained a model to distort the text images to create more difficult examples. In our work, we adopt PBA as the search strategy due to its computation being more efficient than AutoAugment and having a larger parameter space than RandAugment.

## 2.2 LATENT SPACE DATA AUGMENTATION

Interpolation is a simple but effective way to create new data from existing examples. Mixup can operate on general numerical data by interpolating two training examples and their corresponding one-hot encoded labels (Zhang et al., 2018). Manifold Mixup applies Mixup to the outputs from different hidden layers (Verma et al., 2019); LSI applies Mixup in the latent space for image classification (Liu et al., 2018). Previously, Devries & Taylor (2017a) found that extrapolation between features learned by an autoencoder generates more effective synthetic examples than adding Gaussian noise and interpolation. In their experiments, they suspected that interpolation tightens the class boundaries, which leads to overfitting and harms the performance. For text data, Kumar et al. (2019) explored simple latent space transformations and autoencoder-based methods in few-shot intent classification tasks. In general, it remains an open question which transformation in the latent space creates better augmented examples. We leverage automated data augmentation to find the optimal composition of latent space transformations for a variety of datasets of multiple data modalities.

## 2.3 HARD EXAMPLE AUGMENTATION

Kuchnik & Smith (2019) show that not all training examples are equally useful for augmentation and propose methods to find a subset of useful data to augment. Similar methods are also proposed by the recent augmentation algorithms, which aim to find the composition of transformations that create difficult examples (Luo et al., 2020; Wu et al., 2020; Zhang et al., 2020). In our work, we propose a novel and simple way to create hard examples using latent space transformation.

## 3 METHODOLOGY

### 3.1 BRIEF OVERVIEW OF POPULATION BASED AUGMENTATION

We first briefly review the policy search procedure of PBA (Ho et al., 2019). In PBA, each augmentation function is associated with its probability and magnitude, for example, the tuple ($op$: rotation, $p = 0.4$, $\lambda = 0.5$) specifies that the rotation operation is applied with a probability of 0.4 and a magnitude of 0.5. A policy is a set of such operation tuples. When a policy is applied to a mini-batch, 0, 1 or 2 operation tuples are randomly sampled from the policy and applied to the batch of data with their corresponding probability and magnitude.

PBA formulates the augmentation policy search as hyperparameter schedule learning. Instead of a fixed policy, it learns a policy schedule that specifies the augmentation parameters at different stages of training. In particular, PBA exploits population-based training (PBT) to optimize the model weights of multiple child models jointly with their policy parameters that maximize the performance. It starts by simultaneously training a fixed number of randomly-initialized child models, each with its own policy. After certain fixed intervals, the child models are evaluated using the validation set. The worst models clone the weights and policies from the best models. The parameters of the cloned policies are either resampled from all possible values or perturbed from the cloned values. The process repeats until the training of the child models completes. The final output of PBA is a schedule of policies unrolled from the best child model. The policy schedule can be applied to

augment the dataset when training with more data or larger models. As there is no need to re-train any child model, PBA is considered to be an efficient automated augmentation algorithm. Our method follows the policy formulation and the search strategy used in PBA.

### 3.2 MODALS

Our main assumption is that, if learned properly, the region corresponding to a class in the latent space learned by a deep neural network is mostly convex and isotropic. Consequently, linear transformation in such a space would result in a smooth transition from real data to artificial data without altering their class identity. Among other non-trivial properties, data points are often sparsely distributed in high-dimensional spaces such that distance-based methods that work well in lower-dimensional spaces fail to take advantage of the distance metrics (Donoho, 2000). In the supplemental materials, we provide empirical evidence to show that the relatively low-dimensional latent spaces used in our experiments are not anisotropic, especially within the local class regions (see Appendix A.1). In what follows, we introduce the latent space transformations and the training objectives of our model.

#### 3.2.1 TRANSFORMATIONS

**Hard example interpolation.** Given a seed latent representation $z_i^c$, which is the latent representation of the $i$-th example in class $c$, we interpolate $z_i^c$ to a hard example. The hard example is taken as the latent representation nearest to $z_i^c$ among $q$ hard example candidates sampled according to the classification loss. This favours the creation of harder examples and prevents $z_i^c$ from being aggressively interpolated to distant locations or regions that are close to the class center. In our implementation, we take $q$ as 5% of the number of examples in class $c$. Formally, let $S = \{s_i\}_{i=1}^q$ denote the set of hard latent representations sampled according to the magnitude of the loss in class $c$, $s = \arg\min_{s \in S} \frac{s \cdot z_i^c}{\|s\|\|z_i^c\|}$ denote the closest hard example (red circle in Figure 1a), and $\lambda_1$ denote a scaling factor. Hard example interpolation is expressed as:

$$\hat{z_i^c} = z_i^c + \lambda_1(s - z_i^c) \tag{1}$$

**Hard example extrapolation.** The data nearer the class boundaries are more difficult to classify. Instead of extrapolating from random examples, we extrapolate $z_i^c$ from the class center $\mu_c = \frac{1}{m}\sum_{j=1}^m z_j^c$ with $\lambda_2$ as the scaling factor. Hard example extrapolation is thus given by:

$$\hat{z_i^c} = z_i^c + \lambda_2(z_i^c - \mu_c) \tag{2}$$

**Gaussian Noise.** We also perturb $z_i^c$ by adding Gaussian noise with zero mean and per-element standard deviation computed across the examples in the same class. Let $\epsilon \sim \mathcal{N}(0, \sigma_c^2 I)$ and $\lambda_3$ be the scaling factor, we have:

$$\hat{z_i^c} = z_i^c + \lambda_3 \epsilon \tag{3}$$

**Difference Transform.** Difference transform is an alternative way to perturb $z_i^c$ by translating $z_i^c$ along the direction between two random latent vectors sampled from the same class. Denoting the scaling factor as $\lambda_4$, the difference transform can be written as:

$$\hat{z_i^c} = z_i^c + \lambda_4(z_j^c - z_k^c) \tag{4}$$

These four latent space transformations are illustrated in Figure 1.

#### 3.2.2 MODEL

Instead of training an autoencoder to learn the latent space and generate additional synthetic data for training, we train a classification model jointly with different compositions of latent space augmentations. Given an input space $\mathbb{X}$, a latent space $\mathbb{Z}$, and a label space $\mathbb{Y}$, MODALS trains a feature extraction model $F(x; \theta) : \mathbb{X} \to \mathbb{Z}$ and a dense layer to map $\mathbb{Z} \to \mathbb{Y}$. The $i$-th example $x_i$ is mapped to its latent representation $z_i = F(x_i)$ and further augmented by a composition of label-preserving transform functions to obtain the augmented latent representation $\hat{z_i} \in \mathbb{Z}$. We optimize the softmax

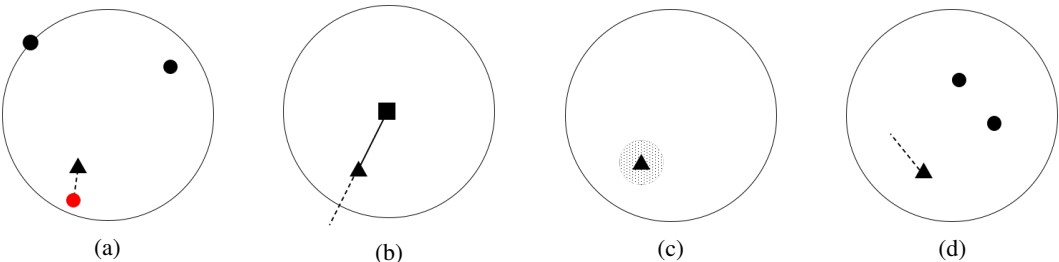

Figure 1: Illustration of (a) hard example interpolation, (b) hard example extrapolation, (c) Gaussian noise, and (d) difference transform (triangle: seed latent representation; circle outline: class boundary; black circle: sampled feature; red circle: nearest sampled hard example; square: class mean; dotted line: range of augmented example)

cross-entropy loss among $k$ classes with $\boldsymbol{W} = [\boldsymbol{w}_1, ..., \boldsymbol{w}_k]$ as the weights of the final dense layer and $M$ as the batch size. The classification loss is given by:

$$L_{clf}(\boldsymbol{W}, \boldsymbol{\theta}) = -\frac{1}{M} \sum_{i=1}^{M} \log \frac{\exp(\boldsymbol{w}_{y_i}^{\top} \hat{\boldsymbol{z}}_i)}{\sum_{j=1}^{k} \exp(\boldsymbol{w}_j^{\top} \hat{\boldsymbol{z}}_i)} \tag{5}$$

In addition to the classification loss, we further employ two additional loss terms to improve the augmentation results. The effect of these two losses will be discussed in Appendix A.2.

**Adversarial Loss.** Sampling in a high-dimensional latent space may fall into invalid regions that are not on the data manifold. In order to produce smooth morphing between sampled latent codes, there are well-studied techniques to enforce a posterior distribution over the latent variables using VAE or GAN-based methods (Kingma & Welling, 2014; Goodfellow et al., 2014). In MODALS, we regularize the latent space by imposing an adversarial loss similar to Adversarial Autoencoder (Makhzani et al., 2016). In particular, we employ a discriminator $D(\boldsymbol{z}; \boldsymbol{\phi})$ to distinguish the latent code generated by the feature extraction model and sampled from the Gaussian distribution. The feature extraction model has to generate latent representations that are similar to the Gaussian distribution to fool the discriminator. This leads to the adversarial objective $L_{adv}(\boldsymbol{\theta})$ for the feature extraction model and the discriminator loss $L_D(\boldsymbol{\phi})$ for the discriminator:

$$L_{adv}(\boldsymbol{\theta}) = -\frac{1}{M} \sum_{i=1}^{M} \log D(\boldsymbol{z}_i) \tag{6}$$

$$L_D(\boldsymbol{\phi}) = -\frac{1}{M} \sum_{i=1}^{M} \left( \log D(\boldsymbol{\epsilon_i}) + \log[1 - D(\boldsymbol{z}_i)] \right); \ \forall i, \boldsymbol{\epsilon_i} \sim \mathcal{N}(\mathbf{0}, \mathbf{I}) \tag{7}$$

**Triplet Loss.** Triplet loss is often used in metric learning to learn the latent representations of different classes (Schroff et al., 2015). Triplet loss pulls together the representations from the same class and repels the ones from different classes. We argue that this characteristic facilitates label-preserving augmentation in the latent space. Intuitively, the data outside the convex hull of the training samples are less likely to correspond to valid data of the same class. As a result of using the triplet loss, the distribution of the latent representations in the same class will be more compact. Interpolating, extrapolating or perturbing latent representations in this denser region is more likely to result in an augmented example sharing the same class identity. In theory, alternative measures, e.g., center loss (Wen et al., 2016), large margin softmax loss (Liu et al., 2016), or other contrastive losses, can also produce similar effects. For simplicity, we use the triplet loss in our implementation: for each anchor latent representation $\boldsymbol{z}$, we randomly sample a positive latent representation $\boldsymbol{z}^+$ from the same class and a negative representation $\boldsymbol{z}^-$ from a different class in the same mini-batch. We denote the margin as $\gamma$ and the cosine distance function as $d(\cdot)$. The triplet loss $L_{tri}(\boldsymbol{\theta})$ is given by:

$$L_{tri}(\boldsymbol{\theta}) = \frac{1}{M} \sum_{i=1}^{M} \left[ d(\boldsymbol{z}_i, \boldsymbol{z}_i^+) - d(\boldsymbol{z}_i, \boldsymbol{z}_i^-) + \gamma \right]_+ \tag{8}$$

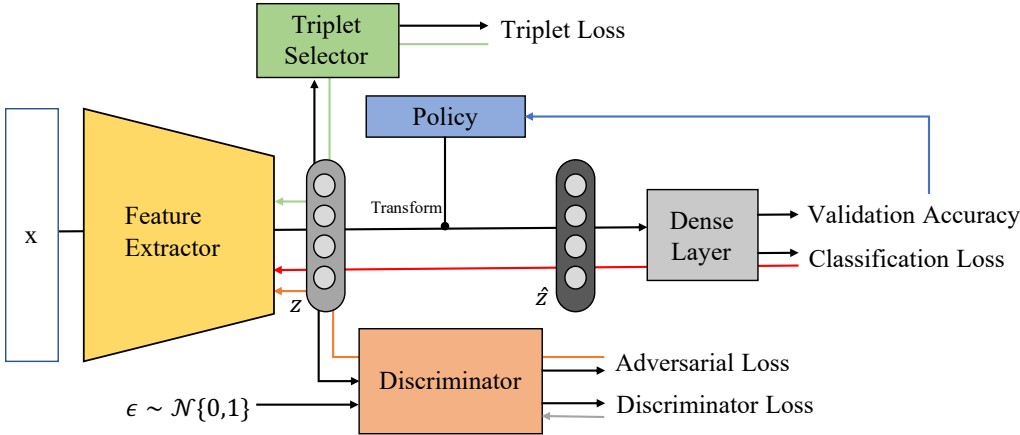

Figure 2: MODALS ($z$: seed latent representation; $\hat{z}$: augmented latent representation; black line: forward propagation; red line: gradient flow from $L_{clf}$; green line: gradient flow from $L_{tri}$; orange line: gradient flow from $L_{adv}$; blue line: reward signal; grey line: gradient flow from $L_D$.)

MODALS is trained by minimizing the combined loss $L = L_{clf} + \alpha L_{adv} + \beta L_{tri}$, where $\alpha$ and $\beta$ are predefined hyperparameters. Figure 2 shows the overall architecture of MODALS : For each mini-batch, a feature extraction model maps the input data to their latent representations. The representations are used to compute the adversarial and triplet losses, and also augmented to obtain the classification loss and validation accuracy. The combined loss $L$ is back-propagated to the feature extraction network while the validation accuracy is treated as the reward signal to update the policy.

### 3.2.3 POLICY SEARCH

We apply the PBA policy search strategy to the latent representations with the four proposed transformations. We define four latent space transformations with probability ranging from 0 to 1 and magnitude $\lambda$ ranging from 0 to 0.9, both in intervals of 0.1. As the same operation can be applied twice, the search space is of size $(10 \times 11)^8 \approx 2.14 \times 10^{16}$. The total computation cost is the cost to train a single model multiplied by the number of parallel models.

## 4 EXPERIMENTS

To demonstrate the modality-agnostic property of MODALS, we evaluate it using datasets from four different data modalities: text, tabular, time-series, and image data. For each dataset, we search for the optimal augmentation schedule using PBT with 16 parallel models perturbing every three epochs. The searched policy schedule is applied to the dataset with different numbers of training examples. We implement the discriminator as a 2-layer multilayer perceptron (MLP) model with 256 hidden units in each layer. In all experiments, we set $\alpha = 1$, $\beta = 0.03$ and search for the metric margin value from $\{0.5, 1, 2, 4, 8\}$. We compare our method with baseline methods that do not involve training an auxiliary model. In particular, we construct the baseline models trained with well-defined input-space augmentation in the data modality and label-mixing augmentation. We report the average classification accuracy from three trials with the baselines. The detailed configuration and policy of each experiment are provided in Appendix A.3.

### 4.1 TEXT DATA

We test MODALS on the SST2 (Socher et al., 2013) and TREC6 (Li & Roth, 2002) datasets, which are popular benchmarks for binary and multi-class text classification tasks. SST2 is a binary sentiment classification dataset with 6,920 training examples, while TREC6 is a question classification dataset with 5,452 training examples from six question classes. We employ a 2-layer bidirectional

LSTM using fixed pre-trained 300-dimensional GloVe as the word embedding (Pennington et al., 2014). The hidden state dimensionality of the model is 256. The first baseline is trained without augmentation. We construct the second baseline by applying manually designed input-space augmentations proposed as easy data augmentation (EDA) techniques in (Wei & Zou, 2019). In this baseline, seed sentences are cloned multiple times, where each cloned sentence is transformed by replacing synonyms, inserting synonyms, deleting random word, or swapping word order. The baseline follows the configurations suggested by EDA for datasets of different sizes. We construct the third baseline using senMixup, which applies mixup augmentation in the last hidden layer before the softmax layer (Guo et al., 2019). We also include the fourth and fifth baselines which use back-translation augmentation. The back-translation baselines are implemented using the Google Translate API to translate the original text to German (DE) or Spanish (ES) and back to English. Then, the model is trained on the back-translated text with the original text. Our experiments show that MODALS outperforms the other baseline methods in five of the settings and is comparable with back-translation (ES) in the TREC6 dataset. While EDA, senMixup and back-translation also show improvement on the full datasets, senMixup and back-translation sometimes slightly underperform the first baseline when the dataset size is small (see Table 1).

Table 1: Comparison of five baselines (training without augmentation, EDA, senMixup, back-translation from German and back-translation from Spanish) with MODALS on two text datasets for different amounts of training examples (s: 10%, m: 50%)

|  | w/o Aug. | EDA | senMixup | BT (DE) | BT (ES) | MODALS |
|---|---|---|---|---|---|---|
| SST2 (s) | 73.01 | 74.52 | 73.27 | 73.49 | 73.04 | **75.67** |
| SST2 (m) | 79.70 | 79.77 | 79.50 | 79.52 | 79.48 | **81.35** |
| SST2 | 81.91 | 82.97 | 82.85 | 82.90 | 83.32 | **84.14** |
| TREC6 (s) | 81.10 | 81.10 | 78.06 | 81.20 | 80.40 | **83.20** |
| TREC6 (m) | 90.44 | 91.13 | 88.67 | 92.47 | 92.27 | **93.00** |
| TREC6 | 92.67 | 93.26 | 93.07 | 93.60 | **94.00** | **94.00** |

## 4.2 TABULAR DATA

We also perform an experiment with multiple tabular datasets from the UCI repository (Dua & Graff, 2017), including the Iris, Breast Cancer, Arcene (Guyon et al., 2005), Abalone, and HTRU2 (Lyon et al., 2016) datasets. The data are standardized and trained with a 2-layer MLP that encodes the inputs to 128-dimensional latent codes. We compare the accuracy with baseline methods using no augmentation and using Mixup. The experiment is repeated with different numbers of training examples. Table 2 shows that MODALS outperforms the baseline methods in 13 out of 15 settings. In the remaining two settings, MODALS achieves the same performance as Mixup augmentation.

Table 2: Comparison of two baselines (training without augmentation and Mixup) with MODALS on five tabular datasets for different amounts of training examples (s: 20%, m: 50%)

| Dataset | w/o Aug. | Mixup | MODALS | Dataset | w/o Aug. | Mixup | MODALS |
|---|---|---|---|---|---|---|---|
| Iris (s) | 86.67 | 86.67 | **96.67** | Abalone (s) | 62.80 | 62.84 | **63.40** |
| Iris (m) | 96.67 | 98.00 | **98.67** | Abalone (m) | 64.04 | 64.67 | **64.95** |
| Iris | 96.67 | **98.89** | **98.89** | Abalone | 64.59 | 65.67 | **65.71** |
| B. Cancer (s) | 95.32 | 95.61 | **96.67** | HTRU2 (s) | 97.78 | 97.85 | **98.10** |
| B. Cancer (m) | 96.78 | 97.08 | **97.37** | HTRU2 (m) | 98.01 | 98.03 | **98.14** |
| B. Cancer | 96.78 | 97.66 | **99.41** | HTRU2 | 98.04 | **98.16** | **98.16** |
| Arcene (s) | 64.33 | 69.67 | **71.00** |  |  |  |  |
| Arcene (m) | 78.00 | 78.00 | **79.33** |  |  |  |  |
| Arcene | 83.33 | 83.33 | **83.67** |  |  |  |  |

### 4.3 TIME-SERIES DATA

For time-series data, we use the HAR (Anguita et al., 2013) and Malware (Catak, 2019) datasets. HAR and Malware belong to continuous and discrete time-series data, respectively. The input of the HAR dataset consists of multiple continuous smartphone accelerometer and gyroscope readings for six human activities. The input of the Malware dataset consists of sequences of discrete events, specifically, the API call sequences of eight malwares. For both datasets, we train an LSTM model to encode the time-series data as 128-dimensional feature vectors and apply MODALS. MODALS performs the best in all cases (see Table 3).

Table 3: Comparison of two baselines (training without augmentation and Mixup) with MODALS on two tabular datasets for different amounts of training examples (s: 10%, m: 50%)

| Dataset | w/o Aug. | Mixup | MODALS | Dataset | w/o Aug. | Mixup | MODALS |
|---------|----------|-------|--------|---------|----------|-------|--------|
| HAR (s) | 49.93 | 52.20 | **57.71** | Malware (s) | 79.24 | 81.14 | **84.07** |
| HAR (m) | 85.44 | 86.05 | **89.06** | Malware (m) | 84.76 | 85.57 | **86.24** |
| HAR | 88.64 | 91.60 | **91.87** | Malware | 86.40 | 87.01 | **87.12** |

### 4.4 IMAGE DATA

For image data, we apply MODALS to CIFAR-10, SVHN, and the reduced versions of these two datasets, as in (Ho et al., 2019). CIFAR-10 and its reduced version contain 50,000 and 4,000 training images, respectively, while SVHN and its reduced version contain 73,537 and 1,000 training images, respectively. We use Wide-ResNet-40-2 as the feature extraction model in all experiments (Zagoruyko & Komodakis, 2016). We compare our method against three baselines. For the first baseline, we apply randomized cropping, horizontal flipping, and color normalization. For the second baseline, Cutout randomly occludes a patch of size $16 \times 16$ from the image on top of the simple augmentation in the first baseline. The third baseline is PBA, which uses the same search strategy as MODALS. Both PBA and MODALS are deployed with their corresponding searched policy schedules on top of Cutout. On the two original and the two reduced datasets, MODALS outperforms simple augmentation and Cutout but underperforms PBA (see Table 4). We suspect that the input-space augmentations used in PBA are able to cover most variations in these datasets due to the continuous nature of image data. As the input-space augmentations are independent of the latent space augmentations, it is possible to combine PBA with MODALS by jointly searching the optimal augmentation policy parameters to further improve the results.

Table 4: Comparison of three baselines (training with simple augmentation, Cutout, and PBA) with MODALS on two image datasets for different amounts of training examples

| Dataset | Simple Aug. | Cutout | PBA | MODALS |
|---------|-------------|--------|-----|--------|
| Reduced CIFAR-10 | 75.62 | 78.90 | **81.00** | 79.36 |
| CIFAR-10 | 92.04 | 92.48 | **92.72** | 92.51 |
| Reduced SVHN | 76.17 | 76.83 | **83.52** | 81.23 |
| SVHN | 96.37 | 96.14 | **96.54** | 96.46 |

## 5 ABLATION STUDY

**Latent space augmentation.** We study the effect of latent space augmentation techniques in MODALS. We remove the latent space augmentation to isolate the effect from applying the triplet loss and adversarial loss. Table 5 summarizes the performance of MODALS across eight datasets from different modalities. Our study shows that the use of latent space augmentation contributes to additional performance gain when trained with different loss objectives. The detailed breakdown is listed in Appendix A.2.

**Additional losses.** We perform an ablation study against the baseline models trained with different loss settings on eight datasets. Both the triplet loss and adversarial loss improves the performance when training with latent space augmentation (see Table 5). In Appendix A.2.1, we provide the details and evidence showing that the triplet loss enforces more compact local class regions to preserve the class labels when applying latent space augmentation.

Table 5: Comparison of average accuracy when trained under different loss settings and augmentation techniques

|  | $L_{clf}$ | $L_{clf} + L_{adv}$ | $L_{clf} + L_{tri}$ | $L_{clf} + L_{adv} + L_{tri}$ |
|---|---|---|---|---|
| w/ Aug. | 86.35 | 88.18 | 88.37 | **89.12** (MODALS) |
| w/o Aug. | 84.71 | 86.64 | 86.74 | 87.69 |

**Hard examples**. We study the effectiveness of our proposed hard augmentation in creating uncertain examples on eight datasets. Taking the difference of the predicted probability between the most likely and second most likely labels as a measurement of prediction certainty, our study shows that the proposed hard example interpolation and extrapolation create 7.23% and 2.34%, respectively, less certain examples on average (see Table 6). We further validate that the augmented examples improve the classification performance on hard examples (see Appendix A.2.2, A.2.3).

Table 6: Comparison of average change in prediction certainty when applied with and without hard augmentation

| w/o Aug. | +**Hard Interpolation** | +**Hard Extrapolation** |
|---|---|---|
| 0.7093 | 0.6581 ($-7.22\%$) | 0.6927 ($-2.34\%$) |

## 6 CONCLUSION

In this paper, we introduce MODALS to apply automated data augmentation in the latent space using four proposed modality-agnostic transformations trained with additional loss metrics and hard example augmentation techniques. Our method is tested on text, tabular, time-series, and image data and can be readily integrated with popular deep learning models. Beyond the data modalities tested, MODALS also work on all other data modalities given proper feature extraction models. We believe that MODALS makes larger improvements to data modalities in which the input space augmentation is less trivial to be defined, like text data, video data or even graph data in low-resource regimes.

ACKNOWLEDGMENTS

This research has been made possible by the Hong Kong PhD Fellowship provided to the first author and research grants (General Research Fund project 16204720 and Collaborative Research Fund project C6030-18GF from the Research Grants Council of Hong Kong; Amazon Web Services Machine Learning Research Award 2020) provided to the second author.

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

## A    APPENDIX

### A.1    VALIDITY OF ISOTROPIC ASSUMPTION IN THE LATENT SPACE

Due to the curse of dimensionality, high-dimensional vectors are usually sparsely distributed. Let us consider a high-dimensional hypersphere with data points evenly distributed inside the hypersphere. As the dimensionality increases, it can be shown that the percentage of data points residing near the surface of the hypersphere will increase significantly. This contradicts our common perception for low-dimensional spaces, such as two- or three-dimensional spaces. Worse still, in case the phenomenon indeed holds, the isotropic assumption about the region of a class will be violated. As such, we want to conduct an experiment to see if there is evidence revealing the occurrence of this unusual phenomenon. Specifically, we study the spread of data points in the latent spaces of different dimensionality used. For each class $c$ of size $m$, we compute the normalized distances $\{r_i^c\}_{i=1}^m$ from the class mean $\boldsymbol{\mu}_c$ to the latent representations $\{\boldsymbol{z}_i^c\}_{i=1}^m$:

$$r_i^c = \frac{\ell_2(\boldsymbol{z}_i^c, \boldsymbol{\mu}_c)}{\frac{1}{m}\sum_{j=1}^m \ell_2(\boldsymbol{z}_j^c, \boldsymbol{\mu}_c)} \tag{9}$$

We repeat the experiment on one dataset for each of the four data modalities. Table 7 reports the average standard deviation of the normalized distances for all classes. The average standard deviation in all datasets shows a decreasing trend as the dimensionality increases, falling between 0.25 to 0.75 for a dimensionality of 128 or 256. In case the unusual phenomenon as described above really

holds so that most of the data points have similar distances from their class center, we would expect the standard deviation to be very small or even close to zero. Our experiment shows that this is not the case especially for lower latent space dimensionality.

Table 7: Average standard deviation of the normalized distances for all classes as the latent space dimensionality $d$ varies

| $d$ | SST2 | TREC6 | CIFAR-10 | SVHN | IRIS | B. Cancer | HAR | Malware |
|---|---|---|---|---|---|---|---|---|
| 2 | 0.7801 | 0.8684 | 0.6068 | 0.6273 | 0.6688 | 0.8229 | 1.0847 | 0.6233 |
| 4 | 0.7592 | 0.7664 | 0.4286 | 0.5458 | 0.6507 | 0.6229 | 1.1415 | 0.4831 |
| 8 | 0.7688 | 0.6725 | 0.4059 | 0.6125 | 0.8674 | 0.6289 | 0.9768 | 0.3983 |
| 16 | 0.7774 | 0.6600 | 0.3789 | 0.5639 | 0.5668 | 0.5267 | 0.8000 | 0.3409 |
| 32 | 0.7812 | 0.6161 | 0.3783 | 0.5496 | 0.5642 | 0.5286 | 1.0814 | 0.3078 |
| 64 | 0.7747 | 0.5743 | 0.3829 | 0.5007 | 0.5526 | 0.4694 | 0.6396 | 0.2793 |
| 128 | 0.7534 | 0.5206 | 0.3799 | 0.4639 | 0.5568 | 0.3329 | 0.8639 | 0.2590 |
| 256 | 0.7297 | 0.4791 | 0.3709 | 0.4681 | 0.5407 | 0.4826 | 0.6127 | 0.2500 |

## A.2 ABLATION STUDY

Table 8: Summary of the performance under different settings (Aug.: latent space augmentation; $L_{adv}$: adversarial loss; $L_{tri}$: triplet loss)

| Datasets | MODALS | $-$ Aug. | $-L_{adv}$ | $-L_{tri}$ | $-L_{adv} - L_{tri}$ |
|---|---|---|---|---|---|
| TREC6 | 94.00 | 92.33 | 93.60 | 93.20 | 91.60 |
| SST2 | 84.14 | 81.36 | 83.54 | 82.28 | 82.01 |
| Iris | 98.89 | 96.67 | 97.78 | 98.89 | 97.78 |
| B. Cancer | 99.41 | 98.83 | 99.41 | 98.83 | 97.36 |
| Reduced CIFAR-10 | 79.36 | 79.40 | 78.88 | 79.33 | 74.68 |
| Reduced SVHN | 81.23 | 80.65 | 79.25 | 78.12 | 74.01 |
| HAR | 91.87 | 89.01 | 91.72 | 90.81 | 90.96 |
| Malware (s) | 84.07 | 83.27 | 82.76 | 83.94 | 82.46 |
| | | | | | |
| Average | 89.12 | 87.69 | 88.37 | 88.18 | 86.36 |
| Change | - | -1.63% | -0.85% | -1.07% | -3.20% |
| (cont') | | $-$ Aug. $-L_{tri} - L_{adv}$ | $-$ Aug. $-L_{adv}$ | $-$ Aug. $-L_{tri}$ | |
| TREC6 | 94.00 | 82.67 | 92.60 | 92.33 | |
| SST2 | 84.14 | 81.91 | 82.15 | 79.52 | |
| Iris | 98.89 | 96.67 | 96.67 | 96.67 | |
| B. Cancer | 99.41 | 96.78 | 97.95 | 97.67 | |
| Reduced CIFAR-10 | 79.36 | 75.62 | 78.06 | 79.02 | |
| Reduced SVHN | 81.23 | 76.17 | 77.72 | 78.07 | |
| HAR | 91.87 | 88.64 | 89.36 | 88.59 | |
| Malware (s) | 84.07 | 79.24 | 79.39 | 81.21 | |
| | | | | | |
| Average | 89.12 | 84.71 | 86.74 | 86.64 | |
| Change | - | -5.20% | -2.75% | -2.87% | |

### A.2.1 LOSS

Here, we study the effect of using additional loss metrics for data augmentation in the latent space. We repeat our previous experiments by removing one or both of the triplet and adversarial losses.

Table 8 shows that adding the proposed loss metrics improves the classification performance across all datasets of different modalities.

We further analyze the effect of the triplet loss in the augmentation context. Using the triplet loss forces the distribution of data points in the same class to be more compact. Consequently, the local class region becomes more convex and isotropic and hence facilitates smoother label-preserving transformation. We validate our conjecture by constraining the latent space to two dimensions to facilitate visualizing the data points (see Figure 3). Without the triplet loss, some class regions overlap and appear in less regular shapes. When the triplet loss is used, the class regions become more compact and appear in more circular shapes as observed from the coordinate axes and the visualization plots. This favors the convex and isotropic assumption in Section 2 of the paper when we propose to apply linear transformation on the feature vectors in local class regions. As the classes are better separated, the transformation can better preserve the original class labels.

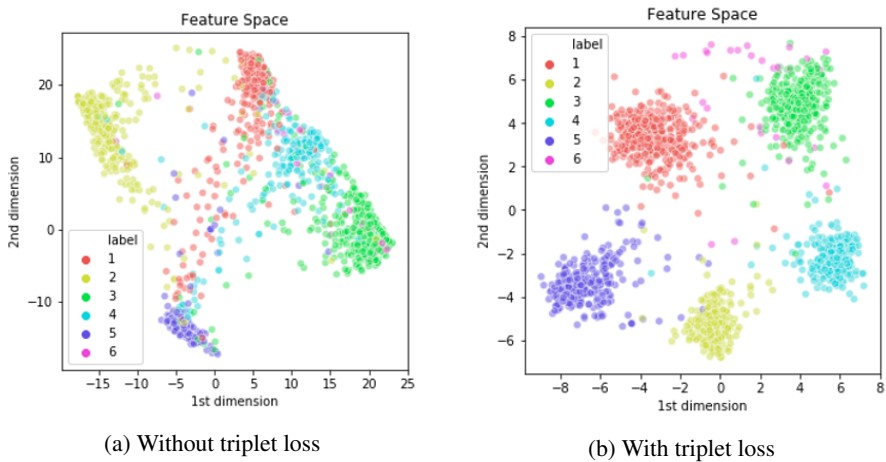

(a) Without triplet loss

(b) With triplet loss

Figure 3: Visualization of data points in the latent space for the TREC6 dataset

In addition to visualizing the data points qualitatively, we also provide quantitative measurements in Table 9 to show the separation between and within the class regions for higher dimensionality. Specifically, we measure the average within-class distance and between-class distance of the latent representations when training with and without the triplet loss. Our study shows that the average within-class distance with the triplet loss is smaller, resulting in a more compact class region. The ratio of between-class and within-class distances suggests a better separation between class regions rather than solely the scaling effect. Similar observations are also noticed in other datasets.

Table 9: Quantitative results for within-class and between-class distances under the effect of triplet loss on the TREC6 dataset. ($d$: dimensionality; $c_w$: within-class distance; $c_b$: between-class distance)

| $d$ | w/o triplet loss | | | w/ triplet loss | | |
|---|---|---|---|---|---|---|
| | $c_w$ | $c_b$ | $c_b/c_w$ | $c_w$ | $c_b$ | $c_b/c_w$ |
| 2 | 9.32 | 25.86 | 2.77 | 1.55 | 7.68 | 4.92 |
| 4 | 6.40 | 20.26 | 3.16 | 1.79 | 7.11 | 3.96 |
| 8 | 7.40 | 21.09 | 2.84 | 2.02 | 7.14 | 3.53 |
| 16 | 8.10 | 19.51 | 2.40 | 2.04 | 6.56 | 3.20 |
| 32 | 9.19 | 22.17 | 2.41 | 2.37 | 7.07 | 2.97 |
| 64 | 9.34 | 22.07 | 2.36 | 2.56 | 7.01 | 2.73 |
| 128 | 9.67 | 21.26 | 2.19 | 3.04 | 7.53 | 2.46 |

### A.2.2 Hard Example Augmentation

Next, we study the effectiveness of our proposed transformation in creating hard examples. In our experiment, we measure the uncertainty of an example by the difference in predicted probability between the most likely and second most likely labels (Scheffer et al., 2001). We call this metric as margin. A smaller margin implies a more uncertain prediction.

$$Margin(\boldsymbol{z}) = p(y_1|\boldsymbol{z}) - p(y_2|\boldsymbol{z}) \tag{10}$$

We train the models on eight datasets and compute the margin for the original features and the augmented features. Our results show that hard interpolation is more effective in creating uncertain examples than hard extrapolation in most datasets (see Table 10). Admittedly, this finding is a bit counter-intuitive. We suspect that it may be due to the different shapes and complexity of the class boundaries, but more investigation is needed to understand the reasons behind.

Table 10: Comparison of the margin between models trained without augmentation and with hard interpolation or extrapolation

| Dataset | w/o Aug. | +hard interpolation | +hard extrapolation |
|---|---|---|---|
| SST2 | 0.4498 | **0.4168** ($-7.33\%$) | 0.4492 ($-0.13\%$) |
| TREC | 0.9681 | 0.9468 ($-2.20\%$) | **0.9273** ($-4.22\%$) |
| Iris | 0.5220 | **0.4884** ($-6.44\%$) | 0.4946 ($-5.25\%$) |
| B. Cancer | 0.8822 | **0.8130** ($-7.85\%$) | 0.8634 ($-2.14\%$) |
| Reduced SVHN | 0.7770 | **0.7066** ($-9.07\%$) | 0.7655 ($-1.48\%$) |
| Reduced CIFAR-10 | 0.8129 | **0.7719** ($-5.04\%$) | 0.7867 ($-3.22\%$) |
| HAR | 0.5069 | **0.4514** ($-10.95\%$) | 0.5044 ($-0.49\%$) |
| Malware (s) | 0.7561 | **0.6697** ($-11.43\%$) | 0.7510 ($-0.67\%$) |

### A.2.3 Hard Example Training

We also study whether the hard examples created using the proposed transformation benefit model training. In the ablation test, we sample the top 5% hardest examples based on the classification loss from each class and compare the changes in the margin and accuracy of the hard examples after training with different augmentation settings (see Table 11). We use a fixed augmentation instead of a fine-tuned policy schedule. In the experiment, the models trained with hard example interpolation and extrapolation achieve a higher accuracy on the sampled hard examples. The larger increases in margin show that the models can learn to predict the hard examples with higher certainty.

Table 11: Comparison of the change in margin and classification accuracy on the hard examples between models trained with different augmentation schemes

| Dataset | w/o Aug. | | +hard interpolation | | +hard extrapolation | |
|---|---|---|---|---|---|---|
| | margin | accuracy | margin | accuracy | margin | accuracy |
| SST2 | 0.0092 | 0.4167 | 0.0719 | **0.4667** | 0.0588 | **0.4667** |
| TREC6 | 0.0981 | 0.0580 | 0.0777 | **0.0592** | 0.0887 | 0.0580 |
| Reduced CIFAR-10 | 0.3572 | 0.1900 | 0.3847 | **0.2533** | 0.3729 | 0.2133 |
| Reduced SVHN | 0.2731 | 0.7926 | 0.3514 | 0.8222 | 0.3441 | **0.8667** |
| Iris | -0.0833 | 0.0370 | 0.1122 | **0.2592** | 0.0231 | 0.0740 |
| B. Cancer | 0.0542 | 0.3589 | 0.1077 | 0.3895 | 0.0666 | **0.4102** |
| HAR | 0.3808 | 0.0895 | 0.4037 | 0.1461 | 0.3530 | **0.1525** |
| Malware (s) | -0.0162 | 0.0044 | 0.0074 | 0.0126 | -0.1121 | **0.0259** |

Finally, we compare the end-to-end training with simple interpolation and extrapolation to our proposed hard augmentation methods. Table 12 presents the accuracy of the model trained with hard augmentation and simple interpolation and extrapolation.

Table 12: Comparison of models trained with simple interpolation and extrapolation and with hard interpolation and extrapolation

| Dataset | Non-hard | Hard |
|---|---|---|
| Reduced CIFAR-10 | 78.11 | **79.36** |
| Reduced SVHN | 80.11 | **81.23** |
| TREC6 | 91.40 | **94.00** |
| SST2 | 82.28 | **84.14** |
| Iris | **98.89** | **98.89** |
| B. Cancer | 96.49 | **99.41** |
| HAR | 88.72 | **91.87** |
| Malware (s) | 81.29 | **84.07** |

## A.3 EXPERIMENT DETAILS

In this section, we present the detailed configuration and augmentation schedule for each of the data modalities. In all the experiments, the augmentation policy is searched using 50% of the data as the validation set. We use 16 child models in PBA implemented using the Ray Tune framework (https://docs.ray.io/en/latest/tune/index.html). The child models are evaluated and perturbed every three epochs. For the discriminator, we employ a 2-layer MLP model with 256 hidden units and ReLU activations. The discriminator is trained using the Adam optimizer with learning rate 0.01.

### A.3.1 TEXT DATA

We employ a single-layer bidirectional LSTM to encode the 300-dimensional GloVe text embedding input as a 128-dimensional latent representation. The model is trained for 100 epochs using the Adam optimizer with learning rate 0.01 and batch size 100. Figure 4(a,b) and Figure 4(c,d) illustrate the augmentation policies for the SST2 and TREC6 datasets, respectively.

### A.3.2 TABULAR DATA

For all tabular datasets, we split 20% of the dataset as the test set unless the test set is explicitly provided in the repository. The tabular data are first standardized and trained with a 2-layer MLP model with 128 hidden units in each hidden layer. We use the Adam optimizer with learning rate 0.01 and batch size 32 to train the model for 30 epochs. In some cases when the dataset size is too small, we use a batch size of 16 instead. Figure 4(e,f) and Figure 4(g,h) show the augmentation policies for the IRIS and Breast Cancer datasets, respectively.

### A.3.3 TIME-SERIES DATA

For the HAR and Malware datasets, the time series are split into subsequences using a sliding window of size 256 with 50% overlap. In the Malware dataset, we remove consecutive duplicated API calls. We use a single-layer LSTM model to encode the time-series input as a 128-dimensional latent representation. For the Malware dataset, we exploit an additional embedding layer to map the discrete API call events into a 32-dimensional input embedding. The LSTM models are trained using the Adam optimizer with learning rate 0.01 and batch size 128 for 50 epochs. The augmentation policies for the HAR and Malware datasets are presented in Figure 4(i,j) and Figure 4(k,l), respectively.

### A.3.4 IMAGE DATA

We use Wide-ResNet-40-2 as the feature extractor for the CIFAR-10 and SVHN datasets. The models are trained using the SGD optimizer with batch size 100 for 200 epochs. We use a weight decay of $10^{-4}$, learning rate of 0.01, and a cosine learning decay with one annealing cycle. Figure 4(m,n) and Figure 4(o,p) show the augmentation policies for the CIFAR-10 and SVHN datasets, respectively.

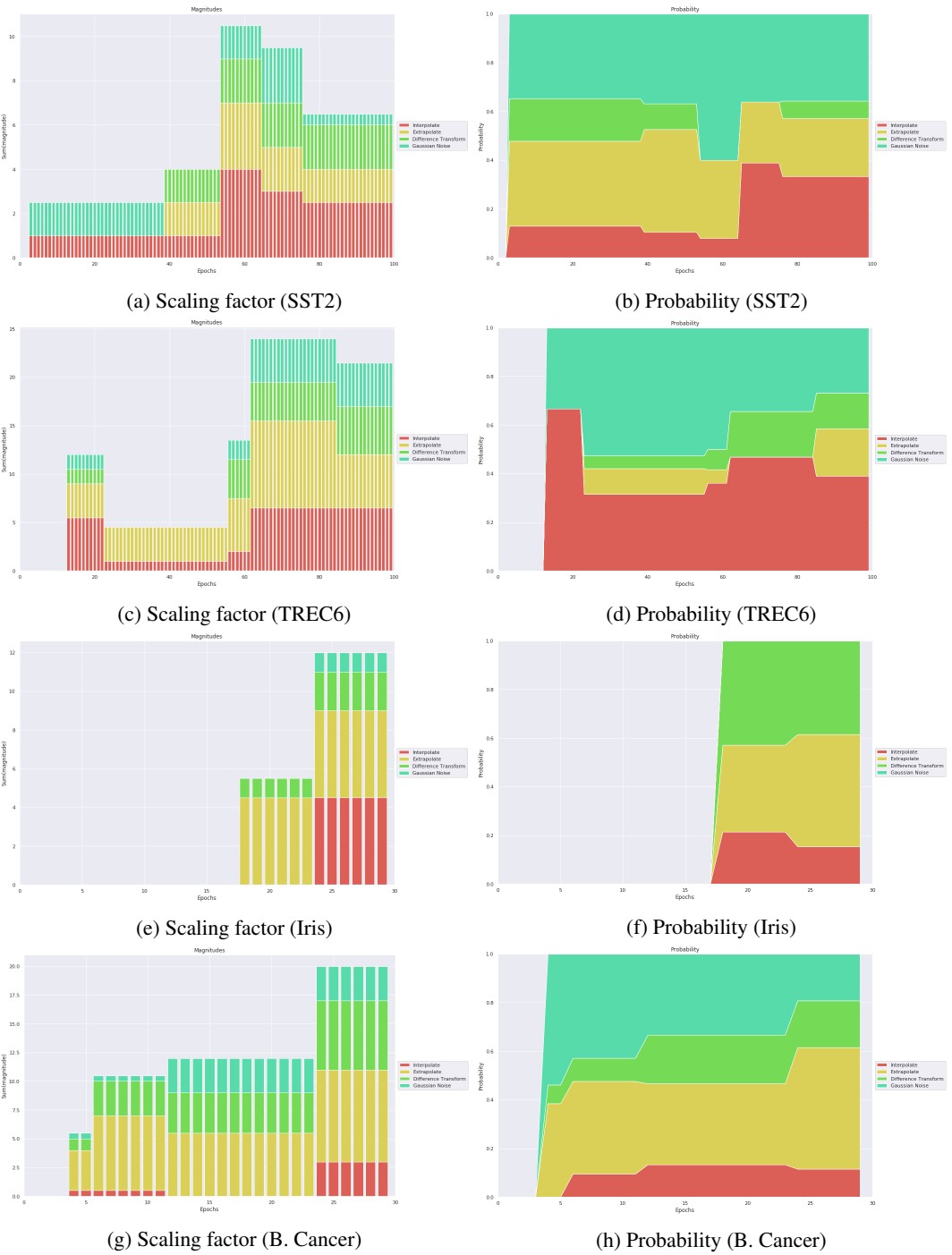

(a) Scaling factor (SST2)

(b) Probability (SST2)

(c) Scaling factor (TREC6)

(d) Probability (TREC6)

(e) Scaling factor (Iris)

(f) Probability (Iris)

(g) Scaling factor (B. Cancer)

(h) Probability (B. Cancer)

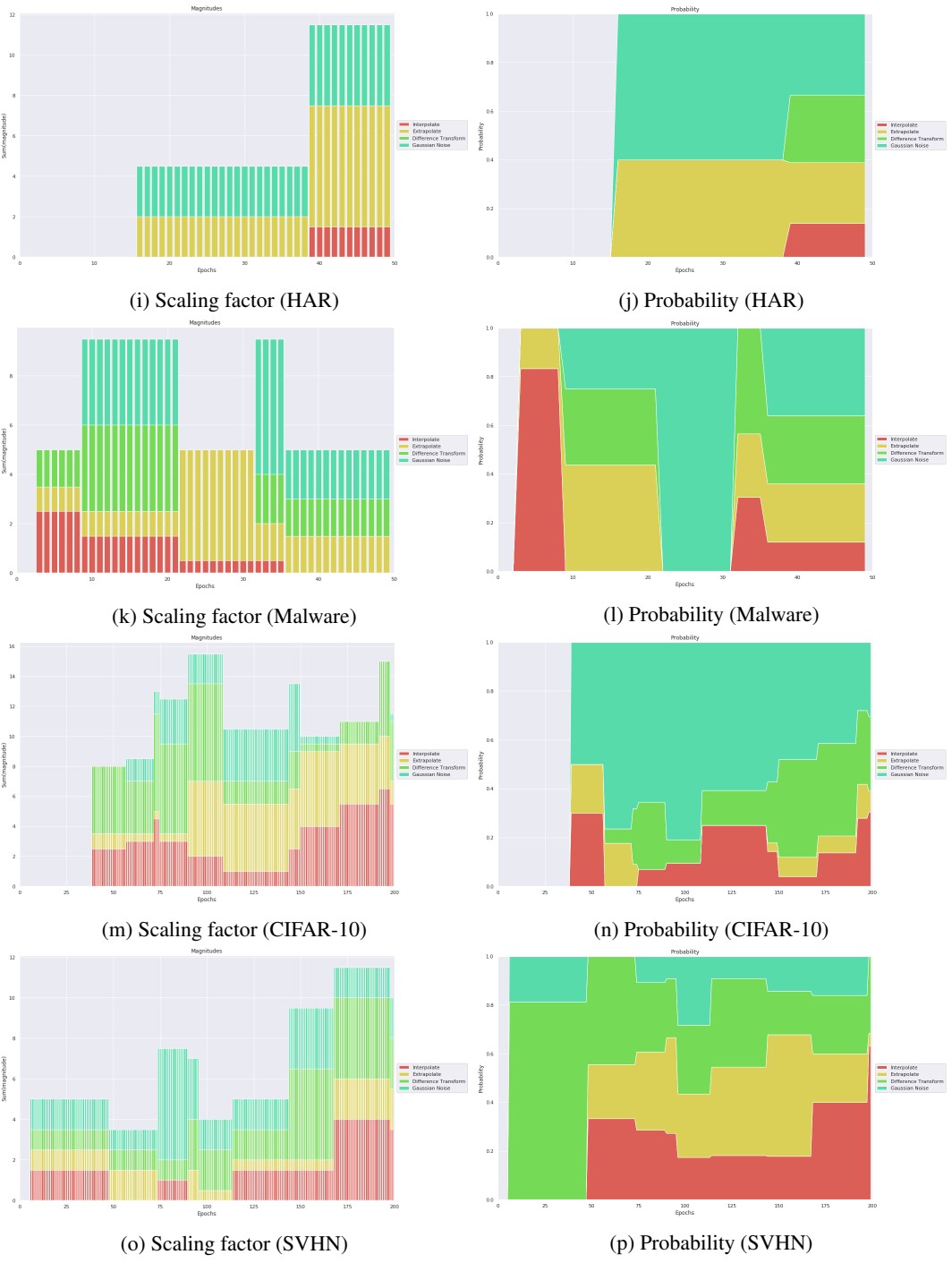

Figure 4: Visualization of the scaling factor and probability of the searched augmentation policies on the SST2 (a,b), TREC6 (c,d), Iris (e,f), B. Cancer (g,h), HAR (i,j), Malware (k,l), CIFAR-10 (m,n), and SVHN (o,p) datasets

