# OpenReview forum: "MODALS: Modality-agnostic Automated Data Augmentation in the Latent Space"
_ICLR.cc/2021/Conference — ICLR 2021 Poster_

### Official Review · AnonReviewer2 · 2020-10-26
**Tackles important problem, consistent improvement, some clarifications required.**

**Rating:** 6
**Confidence:** 3

**Review:**

Summary:
The paper presents MODALS, a new learning objective with automated data augmentation in the latent space. For data augmentation, the paper presents 4 latent-space augmentation primitives by linear interpolation/extrapolation of feature vectors from the same class. For training objective, in addition to multi-class classification objective (formulated as cross-entropy), it introduces two additional losses, such as triplet loss whose triplets are generated using ground-truth class labels and adversarial loss that regularizes feature distribution to be Gaussian. Comprehensive empirical study across multiple domains are presented. The paper demonstrates that the proposed method is particularly effective when the training data is limited.

Pros:
- Data augmentation became very important in deep learning and becomes a bottleneck when applying deep learning methods to problem domains with less prior knowledge on data. In this sense, the paper is tackling very important problem.
- The paper demonstrates consistent performance improvement over baselines across multiple problem domains.
- The presented methods sounds reasonable, though some clarifications are needed how they are evaluated (see below).

Cons/comments:
- While domain-specific data augmentation methods, such as AutoAugment/RandAugment, are typically "class" agnostic, meaning that we don't need to know the class labels of individual data, the presented method requires class label of individual data instances for augmentation. This could be a limitation of the proposed framework, as data augmentation methods become essential for semi-supervised and self-supervised learning methods.
- Are PBA or L_{tri}/L_{adv} used for MixUp training? Since MixUp training also involves hyperparameter (mixing coefficient), it seems fair if mixing coefficient is tuned via PBA. Also, are "w/o Aug" in Table 1-3 using L_{clf} only or L_{clf} + L_{tri} + L{adv} as training objective?
- Can MixUp be used as part of domain agnostic augmentation operations?
- One missing baseline is latent MixUp.
- For PBA, how are validation sets formulated? Is there a reason for other hyperparameters, such as margin, \alpha and \beta, validated separately?
- Figure 3 reminds me of center loss (Wen et al., A Discriminative Feature Learning Approach for Deep Face Recognition) and large-margin softmax loss (Liu et al., Large-Margin Softmax Loss for Convolutional Neural Networks), which pointed out the problem of softmax loss and presented fixes. It would be instructive to discuss some potential of these losses combined with latent space augmentation proposed in this paper.

Reason for rating:
- I believe that the paper tackles an important bottleneck of ML algorithms to be useful in real-world. I tend to recommend for acceptance for initial rating.
- However, there are several concerns that need to be clarified (see Cons) in empirical study of the work.

I have read the author response and have decided to keep my initial positive rating of 6.

---

> ### Author Response · Authors · 2020-11-15
> **Thank you for providing an inspiring perspective and suggesting possible improvements to the proposed MODALS framework.**
>
> __Summary__: Thank you for providing a fascinating perspective and initiating discussions on class-agnostic augmentation and the use of alternative losses. We have updated our submission accordingly.
>
> __Response to Cons 1__: The class-agnostic perspective is inspiring. Indeed, this is also the case for most label-mixing augmentation methods, like Mixup and CutMix. For semi-supervised setting, the problem can be mitigated by introducing pseudo-labels. However, for self-supervised learning, we agree that it is a potential limitation. This can lead to future investigation of class-agnostic latent space transformation to assist self-supervised learning tasks for non-image modalities.
>
> __Response to Cons 2__: $L_{tri}$ and $L_{adv}$ are not used for MixUP or PBA. As MixUp typically performs inter-class interpolation, the triplet loss may not be beneficial as it increases the separation between classes. As the original PBA method does not employ such losses, we keep their policy and apply it without $L_{tri}$ and $L_{adv}$. Also, "w/o Aug" in Table 1-3 uses $L_{clf}$ only. We realize this may raise concerns about whether the performance gain is attributed to the additional losses or the latent space augmentation. In response to this, we have added a new ablation study in an attempt to isolate the effect from the additional loss terms in Table 5 of the latest revised version. It shows that data transformation in the latent space contributes to additional performance gain under different loss settings.
>
> __Response to Cons 3__: The use of triplet loss intentionally promotes more compact local class regions for better inter-class transformation. MixUp, as an inter-class interpolation method, may generate data points far away from the data manifold. Therefore, we believe that MixUp may not be suitable for the current MODALS framework. One viable way is to replace the triplet loss by the consistency loss as a measure to preserve class identity. Then it allows MODALS to incorporate label-mixing augmentation, like MixUp, as a candidate latent space augmentation.
>
> __Response to Cons 4:__: Thanks for your suggestion. In text classification tasks, the senMixUp baseline is similar to latent MixUp. It applies MixUp to the latent sentence embedding. We also tried MixUp on the latent representations in tabular datasets. But we found that it does not work well for shallow model (in our case a 2-layer MLP) and small dataset size.
>
> __Response to Cons 5__: The validation sets are formed following the setting in PBA, which takes half of the training data as the hold-out validation set. We intended to make the policy search comparable to the original PBA, such that both algorithms search for the probability and magnitude in applying augmentation operations. It is possible to add extra parameters $\alpha$, $\beta$ and margin to the hyperparameter search procedure. However, if we allow the weights and margin to vary during training (as it is a hyperparameter schedule search), it may lead to unstable training.
>
> __Response to Cons 6__: Thank you for your suggestion. Both the center loss and the large margin loss are good alternatives to the triplet loss in MODALS. It can potentially be more efficient when the number of anchor pairs in the triplet loss is large. We have revised our submission to mention about using the center loss, large margin loss and other contrastive losses as viable alternatives of the triplet loss.

---

### Official Review · AnonReviewer4 · 2020-10-27
**Good paper, but missing a proper ablation study**

**Rating:** 6
**Confidence:** 4

**Review:**

The paper proposes an automatic data augmentation method that is modality agnostic by modifying the data in a latent space (rather than in input space). They design latent space interventions that yield hard examples (which they claim should improve downstream model learning). They apply population based training on the latent-transformation-policy search (which modifies the latent representation of a classification model directly), on top of training the classification model in question.

The “Hard example interpolation” idea, i.e.: to interpolate towards the 5% hardest examples, is interesting. As opposed to the search and latent aspects of the method, couldn’t this be the cause of any possible gains by the method? A baseline experiment to check would be to use this interpolation method in input-level interpolation augmentations such as MixUp.

The larger concern, however, is that it’s not clear what parts of the method yield good results. For instance, in order to make sure that the latent transformations are being applied in an approximately-gaussian latent, they apply adversarial loss on the latent, to enforce the gaussian prior. They also apply a triplet loss. These are motivated decisions, but it’s hard to verify which of these experimental decisions lead to improvements. The ablation study presented shows the baseline models with the losses added to them, where a more convincing study would show the final MODALS method, with each modification removed one by one. This would not be a big concern if the method yielded much higher performance than the baselines, but it seems to underperform in the image domain (Table 4).

Overall, the method presented is really interesting and seems to work well across a variety of domains. In particular, augmentation has proven to be a hard research direction in text and tabular data, so the fact that this method outperforms the baselines is exciting.

Update after rebuttal: I appreciate the authors' response and the new ablation study. I maintain my original score (marginally above acceptance threshold). I continue to think it's a strength that the method works in many domains, but this strength is slightly diminished due to the variability of domain performance (e.g.: image domain).

---

> ### Author Response · Authors · 2020-11-15
> **Thank you for your constructive feedback. The additional ablation study leads to a better analysis of the proposed model.**
>
> __Summary__: Thank you for your great feedback. We provide an additional scenario in the ablation study (Table 5) to isolate the effect of different factors.
>
> __Response 1__: Thanks for the suggestion. As to isolate the performance gain in sampling hard examples as transformation targets, Table 14 in the ablation study section shows the effect of removing hard example augmentation. Specifically, it shows that the hard augmentation scheme contributes to certain performance gain. For the suggested hard example MixUp baseline experiment: note that our proposed hard augmentations are intra-class transformations, while the standard MixUp operates as inter-class interpolation. If we sample the hardest 5% data across all classes, it may happen that most of the data will be interpolated to a small subset of data.
>
> __Response 2__: Great suggestion. We provide an additional ablation test against each modification, especially for the latent transformations in Table 5 of the latest revised version. It shows that data transformation in the latent space contributes to additional performance gain under different loss settings. Using the adversarial loss and triplet loss also improves the augmentation quality. The detailed breakdown is listed in Table 8 in the Appendix section.

---

### Official Review · AnonReviewer1 · 2020-10-28
**A good generic data augmentation framework, but might need to be compared with hard example generation methods**

**Rating:** 6
**Confidence:** 4

**Review:**

[Summary]
This paper proposes a framework to apply automated augmentation in the latent space which is not restricted to any specific modality. The proposed method, called MODALS, follows the procedure of Population Based Augmentation approach and consists of two main parts: latent space transformation to generate harder examples and auxiliary adversarial loss &   triplet loss to improve augmentation quality. MODALS is evaluated on multiple datasets for text, tabular, time-series and image modalities and achieves higher performance than competitors except on image data.

[Pros]
+ The idea of augmenting data in latent space enables the framework to be generic for any modalities, which has a great potential in applications.
+ This paper is well organized and easy to follow.

[Cons]
- The proposed framework is interesting but somewhat incremental since most of the modules/losses are off-the-shelf.
- The experimental settings seem not very fair and sufficient.
  1. It would be better to equip all the competitors with the triplet loss and conduct the evaluation, to show how the data transformations (Eq.(1-4)) contributes to the learning.
  2. The proposed method is very related to recent hardness aware metric learning method HDML [1] which also propose to manipulate data in latent space to generate hard examples. Could the proposed latent-space automated data augmentation method outperform such metric learning scheme?
- The use of adversarial loss might cause mode collapse, which is  harmful for the data augmentation. How the proposed framework overcome this problem?

[1] Zheng W, Chen Z, Lu J, et al. Hardness-aware deep metric learning. CVPR. 2019: 72-81.

---

> ### Author Response · Authors · 2020-11-15
> **Thank you for your suggestions! Additional comparison has been added to the ablation study.**
>
> __Summary__: We have added one additional scenario to the ablation study to demonstrate how data transformation contributes to learning (Table 5). The additional comparison provides valuable insights for our method.
>
> __Response to Cons 1__: We agree that most of the modules, like the triplet loss and adversarial loss, are existing methods. However, we would like to emphasize that the novelty of our work is to propose a general automated data augmentation framework that can work for multiple data modalities in a generic way. To the best of our knowledge, such an attempt has not been made by others in the research community. As such, the novelty is more on the general framework itself than on the specific techniques used in different components of the framework.
>
> __Response to Cons 2.1__: Thanks for suggesting the comparison. We provide additional experiment results in Table 5 of the latest version. It shows that data transformation in the latent space contributes to additional performance gain under different loss settings.
>
> __Response to Cons 2.2__: Thanks for making us aware of the latent space transformation method in deep metric learning research. Regarding the work of HDML [1], we believe that our method and HDML are applied to different tasks. MODALS is applied to classification tasks while HDML is applied to retrieval and clustering tasks. It would be interesting to see if HDML can also be extended to classification tasks. From the perspective of injecting noise during training, MODALS, as an augmentation method, injects more types of noise (4 types of latent space transformation), while HDML mostly relies on inter-class extrapolation. Consequently, we speculate that MODALS can learn more robust representations. Nevertheless, we think it would be beneficial and viable to include such metric learning technique in MODALS as well. Inspired by HDML, one way to achieve it is to replace the triplet loss by a consistency loss to preserve the class identity and allow inter-class transformations.
>
> __Response to Cons 3__: Empirically, we do not observe mode collapse as a severe issue in MODALS from the experiment results. However, in case some signs of mode collapse are indeed observed, one may reduce the weight $\alpha$ of the adversarial loss to mitigate the problem.

---

### Official Review · AnonReviewer3 · 2020-10-28
**Interesting work on latent space augmentation; Emphasize the need of modality agnostic data augmentation**

**Rating:** 7
**Confidence:** 5

**Review:**

This paper proposes a unified way to augment data in the latent (embedding) space. In particular, the paper combines three existing techniques including adversarial training, triplet loss, and joint training for data augmentation. Paper explores 4 standard modality agnostic data transformations to augment data. Since augmentation is done in the latent space, the proposed technique can be applied to any modality including text, images, time-series data. Empirical results show that the proposed method works better than the standard transformations on all modalities except image datasets.

Strengths:
- The proposed framework works for different kinds of modalities. Paper provides extensive results on various standard benchmark datasets including SST-2, CIFAR-10.
- Paper shows that the examples for which the model is least confident are a good target for the augmentation. This finding is useful for future data augmentation work.
- Paper also provides ablation studies to show the effectiveness of combining adversarial and triplet loss.

Weakness:

Reservations related to the usability of the proposed method: With the results presented in the paper, I am not convinced that the proposed technique will be preferred over other domain-specific data augmentation methods which leads me to question the usability of the proposed model. For example, one can simply use any pre-trained model as in [Kumar et al. 2020] or simply back-translation for text classification instead of the proposed data augmentation and can get much better results. To show the usefulness of this work, it's crucial to show when the proposed method is competitive to the existing strong DA baselines. If authors show that on atleast one datatset, I will increase my score.

- Paper is using weak baseline methods for input space augmentations. For example, for text augmentation, one can either use back-translation which doesn't require any model training and performs better than the EDA baseline used in the paper. Also, MODALS is using a complex architecture that relies on end2end training so I don't think that using a baseline that doesn't require any kind of training is a fair comparison.

- The claim that MODALS's joint training for augmentation is different from previous approaches is not entirely true as the joint training for augmentation has been explored in past. Please refer to [Mounsaveng et al. 2019, Hu et al. 2019] and their citations.

-  Latent space augmentation has been studied extensively in both images and text classification literature. Paper does not provide any comparison with the available latent space methods which seems to work well for a given task. For example, [Kumar et al. 2019] shows a comparison between different latent space augmentation techniques and shows that simply subtracting two sentence embeddings and adding it to the third embedding performs well for the text classification task.

Questions:
-  For hard examples, authors use the difference between the prob of the most likely and the second most likely labels. How does it compare to the simple model confidence baseline where we select the examples for which model is not very confident? Have you done any experiments related to that?
- I am unable to find the implementation details of the discriminator model used in the experiments. Could you please add those details?

References:
-  Mounsaveng, S., Vazquez, D., Ayed, I. B., & Pedersoli, M. (2019). Adversarial learning of general transformations for data augmentation. arXiv preprint arXiv:1909.09801.
-  Kumar, V., Glaude, H., Lichy, C.D., & Campbell, W. (2019). A Closer Look At Feature Space Data Augmentation For Few-Shot Intent Classification. DeepLo@EMNLP-IJCNLP.
-  Hu, Z., Tan, B., Salakhutdinov, R. R., Mitchell, T. M., & Xing, E. P. (2019). Learning data manipulation for augmentation and weighting. In Advances in Neural Information Processing Systems (pp. 15764-15775).


Update after reading the response from the authors:

I would like to thanks to authors for answering my questions and revising the draft. I agree that the main strength of this work is to explore data augmentation methods which are modality agnostic. Since most of the data augmentation research is domain specific, I think this paper will increase awareness for modality agnostic data augmentation methods. I think it's a good paper, hence I am revising my score.

---

> ### Author Response · Authors · 2020-11-15
> **Thank you for your response and suggestion to include stronger baselines.**
>
> __Summary__: Thank you for your suggestions. We provide an additional comparison with stronger text classification baselines. We have updated the submission to clarify the claim of contribution and implementation details of the discriminator.  Your question about other latent space transformation techniques have led to valuable insights.
>
> __Response to Weakness 1__: Instead of improving the model performance in a specific domain or modality, the focus (and also novelty) of this work is to propose a general automated data augmentation framework that can work for multiple data modalities in a generic way. To the best of our knowledge, such an attempt has not been made by others in the research community. We fully understand that taking this approach may not be able to outperform some data augmentations designed for a specific domain. Regarding back-translation, in response to your comment, we have implemented two more baselines using the Google Translate API for German and Spanish. Our experiment results show that MODALS is still a strong method in the SST2 and TREC6 classification tasks (see Table 1). We try to explain the edge of MODALS over off-the-shelf back-translation augmentation from several perspectives: 1) For domain-specific cases, like medical or legal documents, back-translation may not translate well as it may be pre-trained using data from different domains. 2) From the data availability perspective, back-translation models are typically exposed to a large text corpus. 3) Back-translation augmentation is deterministic. It is hard to control the degree of augmentation diversity from pre-built translation models. 4) We believe that MODALS can be applied jointly with back-translation to further improve the performance. For image data, we are investigating to apply MODALS with stronger baselines. We have found that applying the trained MODALS policy with trained PBA input-space policy can generate a 1.49% accuracy gain in the reduced SVHN dataset over the original PBA method.
>
> Also, we agree that MODALS uses a relatively more complicated architecture to search for an optimal augmentation policy. We think that the end2end training effort is crucial to augmentation in datasets from new, less explored domains without much prior knowledge (compared to some other baselines that are designed and validated for specific domains). For the domains that MODALS has been applied to, it is worth to investigate whether the previously searched policy can transfer well to other datasets from similar domains. For example, we search for an optimal policy in general ImageNet dataset and transfer or finetune the policy on downstream image classification tasks. This can potentially reduce the end2end training effort but is limited to explored domains.
>
> __Response to weakness 2__:  Thank you for pointing this out. We would like to clarify that such joint training is less explored in operation-based latent space transformation methods. We have revised our submission accordingly in the latest version.
>
> __Response to weakness 3__: Thanks for making us aware of these relevant previous works. We added the work of [Kumar et al. 2019] as a related work in our latest submission. We understand that specific latent space transformations have been explored for image and text data. In [Kumar et al. 2019], the addition and subtraction of sentence embeddings, called LINEAR, together with their proposed PERTURB and EXTRA methods, are roughly covered in our proposed difference transform, Gaussian noise, and hard example extrapolation. It is also observed that some operations perform better in some datasets but worse in others. We believe that combining different operation-based latent space transformations and searching for the optimal augmentation composition can lead to higher performance gain.
>
> __Response to question 1__:  Yes, in the ablation study, we used the margin measurement. We also tested the simple model confidence baseline and found that the trends and observations are similar.
>
> __Response to question 2__: Thanks for pointing out it. The discriminator is implemented as a 2-Layer MLP model with 256 neurons in each layer with ReLU activations. We optimize the discriminator using the Adam optimizer with learning rate 0.01. These are added to the latest revised version.

---

### Decision · Program_Chairs · 2021-01-07
**Final Decision**

**Decision:**

Accept (Poster)

**Comment:**

This paper proposes a unified way of data augmentation using a latent embedding space --- it learns a continuous latent space for transformation, and finds effective directions to traverse in this space for data augmentation. The proposed approach combines existing approaches for data augmentation, e.g., adversarial training, triplet loss, and joint training.  The paper also identifies input examples where the model had low performance and creates harder examples that help the model improve its performance. It is evaluated on multiple corresponding to text, table, time-series and image modalities and outperforms SOTA except on image data.

The paper has responded to the reviewers' feedback to provide more detailed experiments with stronger baselines and also ablation studies to show the effectiveness of different components of the approach. The results can be further improved by thorough empirical comparison to other SOTA methods, and by using other loss functions (e.g.,center loss, large margin loss and other contrastive losses) as alternatives to the triplet loss proposed in the paper.

Some reviewers have pointed out that the paper is somewhat limited in it's novelty, since it combines existing off-the-shelf modules/losses and similar methods have been tried in the past --- the novel contributions of the paper should be clearly highlighted in the revised submission.